# Mathematical Modeling of the Evolution of Absenteeism in a University Hospital over 12 Years

**DOI:** 10.3390/ijerph19148236

**Published:** 2022-07-06

**Authors:** Luc Vialatte, Bruno Pereira, Arnaud Guillin, Sophie Miallaret, Julien Steven Baker, Rémi Colin-Chevalier, Anne-Françoise Yao-Lafourcade, Nourddine Azzaoui, Maëlys Clinchamps, Jean-Baptiste Bouillon-Minois, Frédéric Dutheil

**Affiliations:** 1Occupational and Environmental Medicine, CHU Clermont-Ferrand, F-63000 Clermont-Ferrand, France; lvialatte@chu-clermontferrand.fr; 2Biostatistics Unit, CHU Clermont-Ferrand, F-63000 Clermont-Ferrand, France; bpereira@chu-clermontferrand.fr; 3LMBP, Mathematics, Université Clermont Auvergne, F-63000 Clermont-Ferrand, France; arnaud.guillin@uca.fr (A.G.); anne.yao@uca.fr (A.-F.Y.-L.); nourddine.azzaoui@uca.fr (N.A.); 4Cikaba, F-63000 Clermont-Ferrand, France; miallaret@cikaba.com; 5Centre for Health and Exercise Science Research, Department of Sport, Physical Education and Health, Hong Kong Baptist University, Hong Kong; jsbaker@hkbu.edu.hk; 6CNRS, LaPSCo, Physiological and Psychosocial Stress, CHU Clermont-Ferrand, WittyFit, Université Clermont Auvergne, F-63000 Clermont-Ferrand, France; remi.colin@doctorant.uca.fr (R.C.-C.); mclinchamps@chu-clermontferrand.fr (M.C.); jbbouillon-minois@chu-clermontferrand.fr (J.-B.B.-M.)

**Keywords:** absenteeism, hospital, sociodemographic factors, occupational factors, statistical model

## Abstract

Increased absenteeism in health care institutions is a major problem, both economically and health related. Our objectives were to understand the general evolution of absenteeism in a university hospital from 2007 to 2019 and to analyze the professional and sociodemographic factors influencing this issue. An initial exploratory analysis was performed to understand the factors that most influence absences. The data were then transformed into time series to analyze the evolution of absences over time. We performed a temporal principal components analysis (PCA) of the absence proportions to group the factors. We then created profiles with contributions from each variable. We could then observe the curves of these profiles globally but also compare the profiles by period. Finally, a predictive analysis was performed on the data using a VAR model. Over the 13 years of follow-up, there were 1,729,097 absences for 14,443 different workers (73.8% women; 74.6% caregivers). Overall, the number of absences increased logarithmically. The variables contributing most to the typical profile of the highest proportions of absences were having a youngest child between 4 and 10 years old (6.44% of contribution), being aged between 40 and 50 years old (5.47%), being aged between 30 and 40 years old (5.32%), working in the administrative field (4.88%), being tenured (4.87%), being a parent (4.85%), being in a coupled relationship (4.69%), having a child over the age of 11 (4.36%), and being separated (4.29%). The forecasts predict a stagnation in the proportion of absences for the profiles of the most absent factors over the next 5 years including annual peaks. During this study, we looked at the sociodemographic and occupational factors that led to high levels of absenteeism. Being aware of these factors allows health companies to act to reduce absenteeism, which represents real financial and public health threats for hospitals.

## 1. Introduction

Absenteeism is a very accurate barometer of the malfunctioning of an institution and of the quality of working conditions [1]. Absenteeism has a triple cost: financial, organizational, and cultural [2]. In hospitals, it can represent a threat to the balance of work teams [3] and can alter the quality of care [4]. Even if there have been several definitions of absenteeism since the issue was first raised by Mayo, a renowned psychologist in the 1920s [5,6], the most common definition of absenteeism is the one from the National Agency for the Improvement of Working Conditions (NAIWC), which states: “Absenteeism characterizes any absence that could have been avoided by a sufficiently early prevention of the factors of deterioration of the quality of work conditions, understood in the broadest sense” [7]. This definition suggests that the phenomenon is avoidable and can be reduced by prevention. Indeed, it is an indicator of the state of health of employees and their personal problems. It is also a marker of the organizational situation of hospitals and allows us to question the place that work occupies in society [5]. In 1995, Forssman quickly categorized absenteeism according to different factors, including illness, accidents, leave, and unauthorized or unjustified absences [8]. Absenteeism can be the result of high workloads, aging of the workforce, and increases in the legal retirement age [5]. A recent analysis by the Ministry of Labor reveals that the probability of an agent’s absence depends on sociodemographic variables such as age, gender, marital status, or socio-professional status [9]. When employees are absent, it often creates a burden on the company, not only for the loss of productivity [10] but also for those who are responsible to complete the work neglected by the sick employee [11]. Short-term absences are the most penalizing [12] because they lead to a reluctance to replace absentees and a short-term replacement may not be as good as an incumbent. They create “micro-failures” in the organization [13]. Increasing sharply in the 2000s [13,14] and particularly important in the health field [5,6,12,15,16], these reached an alarming rate in 2017 by being the sector with the greatest sick leave [17]. The average absenteeism rate in hospitals in 2019 was between 8.5 and 10% versus 5.11% in general, according to the French Hospital Federation. This statistic does not only affect medical staff; between 2014 and 2017, absences of hospital employees, excluding doctors, increased from 23.3 to 26 days per year on average [18]. Many articles deal with the financial dimension of absences [14], explaining, in particular, the growing awareness of the costs of presenteeism and absenteeism [19]; but none of the articles deals with a large volume of agents followed over the long term (several consecutive years) and integrating a minimum of information on professional and sociodemographic characteristics.

Therefore, our objectives were to understand the general evolution of absenteeism in a university hospital from 2007 to 2019 and to analyze the professional and sociodemographic factors influencing this problem.

## 2. Methods

### 2.1. Study Design

We conducted a longitudinal study on a data set listing all the starts of absences of the agents of the University Hospital of Clermont-Ferrand with their characteristics from January 2007 to December 2019. This study obtained the agreement of the Ethical South-East VI Committee, the Protection of Individual Rights, and the National Commission for Information Technology and Civil Liberties (CNIL). In agreement with the CNIL, a unique six-digit identifier was assigned to each agent to ensure the anonymity of the data. 

### 2.2. Study Population

No exclusion criteria were applied. The only inclusion criterion was being a staff member of the University Hospital of Clermont-Ferrand, regardless of profession, field, or site.

### 2.3. Judgment Criteria

We conducted an initial study of all types of absences other than annual leave, including absences for training, strikes, union activities, maternity, commuting accidents, work-related accidents, ordinary sick leave, long-term sick leave, occupational illness, family events, authorization for absence, unauthorized absence, unpaid absence, and miscellaneous absences. Then, we inserted the notion of time into the analysis by transforming our data set into a time series. For each absence, we had the start date, the end date, and the duration of the absence. In addition, we had sociodemographic and professional variables to provide further detail. The sociodemographic variables were age, gender, marital status, whether the individuals were parents, the age of the youngest child, and the distance between the absent person’s home and their place of work. The professional variables were the field in which the absent person works, divided into four categories (medical, paramedical, administrative, technical). Other variables were the site at which the absent person worked, whether the person had tenure, and whether the person was a caregiver. 

### 2.4. Statistics

Categorical parameters were expressed as frequencies and associated percentages, the average of the monthly absence rates, and the median and interquartile range (IQR) i.e., 25th and 75th percentiles were given for each modality. The assumption of normality (Gaussian) distribution was evaluated (a). Firstly, according to epidemiological relevance and statistical distribution, certain variables had to be grouped into classes (b). We grouped age into four classes (<30 years, 31–40 years, 41–50 years, >50 years), age of youngest child into four classes (no child, 0–3 years, 4–10 years, >11 years), distance into two classes (<12 km, >12 km), and duration of absence into six classes (1 day, 2–7 days, 8–30 days, 31–90 days, 91–180 days, 181–365 days). We began by examining the proportions of absences for each variable. This means we compared, using bar plots, the monthly averages of the proportions of absences from our 13 years of study of the modalities of each variable to obtain a sense of the factors that most influence absence (c). Then, each modality was compared to know the significance of the differences between them (d). This allowed us to target the factors that were the most important in our study and to establish an initial profile of absent people. We then examined how the different factors changed over time. We transformed our data set into a monthly time series so that each row corresponded to a month. We then had the proportion of absences for each modality by month. This allowed us to analyze and compare the evolution of the modalities of each variable from 2007 to 2019 by plotting them against time. Subsequently, the curves were smoothed annually in order to look for potential break points and to be able to perform linear regressions before and after these breaks (e). The next step was to gather related variables to develop profiles of absentees. To accomplish this, we performed a principal component analysis on our time series, treating months as individuals [20,21] (f). The goal was to find principal axes that would summarize all the variables with as little information missing as possible. PCA would also allow us to study the relations between the modalities of the variables and determine groups of patients sharing very similar characteristics. Each variable was represented by its projection using coordinates on a subspace generated by the PCA axes. The analysis also offered a quality of representation for the variables defined by the squared cosines of the angle it formed with its projection. Indeed, the higher this value is, the better the variable will be represented and will have an importance in the PCA. Finally, a contribution was attributed to each variable defined by the quality of representation multiplied by 100 and divided by the total of the qualities of representations of the component. All this depended on the selection of the number of axes, which must allow us to summarize and extract the most information possible in a few dimensions. To accomplish this, we used the “elbow” method, which consists of finding an abrupt change on the graph presenting the decrease in the eigenvalues measuring the quantity of variance explained by the axis (g). The mathematical details are shown in Appendix A [22]. First, we needed to replace the missing values of the geriatric and rehabilitation department and the secondary site variables that would distort the PCA. Indeed, the curves of their proportions of absences were different from the others and concentrated all the information of the second axis (h). Following this, we realized our PCA and, by looking at the variables that contribute the most to each axis, we were able to discern different profiles. The results were presented in the form of a circle of correlations of the variables and their contributions to the selected axes. It was also interesting to divide our 156-month period into three and compare the profiles obtained with a PCA for the first 52 months, then the next 52, and finally the last 52. A sensitivity analysis was then carried out by taking periods of 50 months and 54 months. This allowed us to see if some variables had more weight than others over specific time periods. Finally, we extracted the principal components found and displayed them as a function of time, which allowed us to integrate them into a vector autoregressive (VAR) model [23,24]. A specificity of this multivariate time series model is to predict a series according to its past data but also according to the past data of other series. Indeed, the model uses an equation with a matrix of constants, matrices of coefficients for each past value, matrices of past values, and a matrix of independent white noise whose mean is equal to 0 to calculate the current values. It thus predicts future values with this system. The number of past values taken into account can be selected when creating the VAR model. The mathematical details are shown in Appendix A [24]. Several tests were performed to ensure the stability and accuracy of the VAR model (i). Predictions were made from the first 130 months for the next 26 months. Indeed, the first 130 months were used as a training period to make our predictions, which we compared to our test period, months 131 to 156. This allowed us to compare the predictions of our model with the actual values and ensure accuracy. The results were expressed using graphs showing the predictions of the two principal components extracted, one representing the profile of people with the highest proportions of absences, the other of people with the lowest proportions. We could then determine the evolution of the absentee profiles in the years to come through predictions (j).

Statistics were performed using RStudio (v 4.1.2, US) and Stata (v16, StataCorp, College Station, TX, USA) software packages. The hypothesis of normality of the distribution was verified using the Shapiro–Wilk test and the function “shapiro.test()” on the RStudio software (a). The grouping of numerical variables into categorical variables with several classes was accomplished with the “recode” function on Stata (b). The bar plots were realized with the functions “ggplot()” and “geom_bar()” of the package “ggplot2” of RStudio (c). Significant differences between the modalities were calculated using the Student’s t-test in order to compare two means. The “t.test()” function was used (d). To determine the break points of them, we had to smooth the curves by taking annual rather than monthly absences with the “apply.yearly()” function. Then, general trends emerged; we were able to compare the modalities of each variable with the other variables visually and through the coefficient of the regression lines (e). We performed the temporal PCA with the “prcomp()” function. This type of PCA will automatically consider the correlations that exist between the months since we base it on time series of frequency 12 (f). The percentage of variance explained by the axes is represented by a graph of the “fviz_eig()” function. The qualities of representations (squared cosines) are expressed with the function “fviz_cos2()” and the contributions with “fviz_contrib()” (g). The “na.approx()” function in the “zoo” package of R was used to perform a linear interpolation of missing data for the geriatric and rehabilitation department in 2010. We performed a linear regression between the last known value before the NA and the first known value after the NA. For the secondary site with missing data from 2007 to 2010, the “imputePCA()” function in the “missMDA” package was more appropriate because we did not just have 1 year without data in the middle of our data set but 3 years of absences at the beginning. The function automatically replaced these values according to the others so that the shape of the curve corresponded to the proportions of absences if the site was open (h). The “VAR()” function of the “vars” package on R allowed us to code the VAR model. Thanks to the “season” parameter, we could specify to our model the presence of periodicity in our data. The Dickey–Fuller test allowed us to check if a time series was stationary, i.e., if its statistical properties did not vary over time and, therefore, if it could be studied. It was made with the “adf.test()” function on RStudio. The output of the model included a *p* value and an adjusted r-squared that measured the percentage of variance explained by it. These values told us whether our model predicted values that were close to reality. A Portmanteau test was performed with the “serial.test()” function to observe the presence or not of autocorrelation; a multivariate Lagrange multiplier ARCH test (“arch.test()” function) was performed to show the presence or not of heteroscedasticity, which could distort our model. Finally, the Granger causality test allowed us to know if the past values of a time series were useful in predicting those of another series (i). Model predictions were made with the “predict()” function (j).

## 3. Results

### 3.1. Description of the Population

The study covered 13 years. There were 1,729,097 absence starts for 14,443 different agents. There were 73.8% women, 55.1% tenured employees, and 74.6% caregivers. Approximately 46.1% worked in the paramedical field, 23.3% in the medical field, 15.1% in the administrative field, and 15.5% in the technical field. Overall, the number of monthly absences was increasing and evolving logarithmically over the period (Figure 1).

### 3.2. Descriptive Analysis

Based on the initial analyses, some variable modalities had higher proportions of absences. We detailed the factors influencing absences variable by variable. Firstly, concerning the sociodemographic variables (Figure 1), women were more absent than men (*p* value = 0.03). Indeed, there was a monthly average of 18.3% with a median at 18.3 and an IQR from 15.1 to 21.9 of absent women compared to 16.6% for men (median at 16.7, IQR 12.8 to 19.6). The age group with the most absences was 30–40 years old with an average monthly absence rate of 21.8% (median 22.3, IQR 18.4 to 26.8) followed by people between 40 and 50 years old (19.3%, median 20.2, IQR 15.5 to 23.6). When comparing each age group in pairs, the absence rate for those between 30 and 40 years of age was significantly different from all other rates (*p* value (30–40 years) vs. (<30 years) < 2.2 × 10^−16^, *p* value (30–40 years) versus (40–50 years) = 0.005, *p* value (30–40 years) vs. (>50 years) < 2.2 × 10^−16^). The proportion of absenteeism of people in couples was the highest of the different marital statuses at 18.9% (median 19.2, IQR 15.8 to 22.6) followed by separated people (17.7%, median 17.5, IQR 14.3 to 21.5), and, finally, single people (14.2%, median 14.4, IQR 11.3 to 17.4). In comparison, only the coupled and separated classes were not significantly different (*p* value = 0.11). The coupled and single classes had a *p* value of 1.39 × 10^−11^, and the single and separated classes had a *p* value of 3.49 × 10^−7^. Distance did not particularly influence absences. Nevertheless, we noted that the average proportion of absences per month was slightly higher for people living more than 12 km from work (18.9%, median 19.1, IQR 15.5 to 22.5) than for people living nearby (16.8%, median 17.1, IQR 13.8 to 20.2). The *p* value of the difference between the two was 0.01. Regarding the children, parents were more likely to be absent (19.1%, median 19.7, IQR 15.6 to 22.9) than people without children (14.2%, median 14.3, IQR 11.5 to 17.1) (*p* value = 7.81 × 10^−12^); we found a higher proportion of absentees among people with young children (21.8%, median 21.7, IQR 18.5 to 26.1 for people with a youngest child under 4 years old; 21.5%, median of 22.1, IQR 17.5 to 26.7 with a youngest child between 4 and 11). The results of the socio-professional variables were as follows: the geriatric and rehabilitation department had the highest average proportion of absences among the different sites (20.2%, median 20.2, IQR 15.0 to 23.9). Next, was the main site (18.0%, median 13.2, IQR 14.9 to 21.1) and then the secondary site (11.5%, median 18.3, IQR 5.0 to 18.2). The pairwise differences were all significant (*p* value primary site versus secondary site = 5.81 × 10^−14^, *p* value primary site versus geriatric and rehabilitation service = 0.01, *p* value secondary site vs. geriatric and rehabilitation service < 2.2 × 10^−16^). The average proportion of tenured staff was higher than that of non-tenured staff (18.4%, median 19.1, IQR 15.2 to 22.2 versus 13.4%, median 13.5, IQR 11.0 to 15.7) (*p* value = 7.85 × 10^−14^). The same was true for caregivers, who had an average absence rate of 18.5%, median of 19.1, and IQR of 15.2 to 22.2, compared to 16.0%, median 15.6, and IQR 12.3 to 18.9 for non-caregivers (*p* value = 0.001). We noted a slight domination of the paramedical field (19.0%, median 19.7, IQR 15.8 to 23.4 against 16.5%, median 15.9, IQR 11.5 to 20.2 in the administrative field; 16.2%, median 15.9, IQR 11.7 to 19.7 in the medical field; and 14.9%, median at 14.3, IQR 11.2 to 17.4 in the technical field). The significant pairwise differences were medical vs. paramedical (*p* value = 1.75 × 10^−4^), paramedical vs. administrative (*p* value = 0.004), and paramedical vs. technical (*p* value = 8.27 × 10^−8^).

### 3.3. Time Series

Next, we examined our data set as a time series to understand how the modalities of each variable changed over time. Plotting them graphically, we noticed that all modalities showed annual peaks. (Figure 2) These were periodic time series. The proportions of absences did not particularly increase; indeed, for most of the modalities, they stagnated around the average. Looking at the linear trend over time of each variable, a certain group of factors could be identified that had almost the same break point around 2011 and had since shown a slight increase in the proportion of absences. These were the only factors showing growth, on average, by +1.5% over the 2011–2019 period. This was the case for age groups 2 and 3, for women, people in couples, separated people, parents, people with a youngest child between 0 and 4, and tenured staff. The variables with the lowest proportions of absences were those with a declining trend for most of them. Among them, age group 4, single persons, persons without children, persons living less than 12 km from the UHC, non-caregivers, and the technical field had no or only one break point in 2009 and an average decrease of 1.5% in the proportion of absences. That decrease reached 5% for men over the 13 years of study. The example of the gender variable is shown graphically in Figure 3. 

### 3.4. Principal Components Aanalysis

These different curves led us to group the modalities with the same evolution since 2007 to find profiles of absentees as a function of time. We, therefore, performed a principal component analysis on the months. PCA allowed us to identify the links between variables based on their correlations and to condense the information into a reduced number of axes. We could then graphically visualize the distance between the variables and, thus, their similarity. We retained two axes for our analysis as suggested by the elbow method on the eigenvalue graph (Figure 2). This explained 91.2% of the information, which was sufficient. It was also noted that the first axis contained almost all the information (more than 87.8%). We then observed the contributions of the variables to the axes, thanks to the correlation circle (Figure 4). In fact, the variables that contributed the most were the factors that had the greatest influence on the evolution of absences in each dimension. According to the contribution graph, the variables that contributed most to axis 1 were having a youngest child between 4 and 10 years old (6.4% of contribution), being between 40 and 50 years old (5.5%), being between 30 and 40 years old (5.3%), working in the administrative field (4.9%), being tenured (4.9%), being a parent (4.9%), being in a couple (4.7%), having a youngest child over the age of 11 (4.4%), and being separated (4.3%). We found the factors that most influenced absences were observed previously in our first analysis. These were the variables that represented an almost identical proportion of absences, and all had the same trend over time. For the second axis, we found that absences in the medical field concentrated a lot of information (32.1% of contribution). We then had the administrative field (9.8%), the geriatric and rehabilitation department (7.8%), non-tenured staff (7.4%), non-health care workers (7.3%), people under 30 years of age (7.1%), the technical field (4.5%), and people over 50 years of age (4.1%). Overall, we found the factors with the lowest proportions of absences. The presence of the administrative field and the geriatric and rehabilitation department could be explained by their significant volatility over the period studied. A sensitivity analysis showed that the absence of these two variables in the PCA did not change the largest contributions to the axes. Similarly, a sensitivity analysis was performed by removing the “secondary site” variable and showed no significant difference in the PCA results. These principal components were then plotted against time. The first component had annual peaks at 50 with a peak of up to 100 in 2009, characterizing the highest proportion of absence for the typical absentee profile, while the second component peaked at 20, synonymous with a factor profile with much lower proportions of absences.

### 3.5. Principal Component Analysis by Period

It was then interesting to see if the weight of each variable changed over time and if the profiles evolved. We then decided to separate our data set into three different periods to observe and compare the profiles of each. We had a 156-month data set, which was separated into three 52-month periods. We then performed a PCA on each time interval. The results of the first PCA, from January 2007 to April 2011, suggested that two axes should be retained. We chose to focus only on the first axis to compare the profile of the most absent among them according to the period. The largest contributors to axis 1 were those whose youngest child was between 4 and 10 years old (6.5% of contributions), those working in administration (5.6%), those between 40 and 50 years old (5.6%), tenured staff (5.3%), those between 30 and 40 years old (5.0%), parents (5.0%), those with a youngest child over 11 years old (4.9%), non-caregivers (4.9%), those in couples (4.8%), and the oldest people (>50 years old) (4.6%). The variables that were major contributors to axis 1 in the overall PCA were found. We noted only the presence of the oldest people and non-caregivers in the top 10 contributions of the first axis over the first period. In the second PCA, from May 2011 to August 2015, the results obtained were similar to those in the first period PCA. Indeed, the largest contributions of axis 1 were having a youngest child between 4 and 10 years of age (6.7% of contributions), being between 30 and 40 years of age (5.6%), working in administration (5.2%), being between 40 and 50 years of age (5.1%), being a parent (4.7%), being a tenured employee (4.6%), being in a couple (4.4%), being separated (4.2%), working in the paramedical field (4.1%), and living more than 12 km from work (4.1%). The differences to be noted, compared to the first period, were the appearance in the largest contributions of absences of separated persons, of those in the paramedical field, and of persons living more than 12 km from work. This meant that their proportions of absences were greater in the second period. Finally, the third PCA, from September 2015 to December 2019, confirmed the pattern found since the beginning of our analyses. In other words, the greatest contributions of axis 1 were having a youngest child between 4 and 10 years of age (5.9% of contributions), being between 40 and 50 years of age (5.5%), being between 30 and 40 years of age (5.2%), being part of a couple (4.7%), being a parent (4.7%), being a tenured employee (4.3%), and working in the administrative field (4.0%). However, there were new factors influencing absences recently. Indeed, having a youngest child between 0 and 3 years old (4.6%), being a woman (4.2%), and being in the medical field (4.0%) contributed much more to the last period than in the previous two. We could, therefore, deduce from these PCAs that a group of seven factors was always present in the top 10 contributions regardless of the period. Sensitivity analysis showed identical results with periods shorter or longer than 2 days.

### 3.6. Autoregressive Vector

Finally, we tried to predict the evolution of these profiles in the years to come. It was, therefore, necessary to use a multivariate time series model. We, therefore, opted for an autoregressive vector. Indeed, this type of model permits us to predict the values of a time series according to its past values but also according to the past values of other time series. We used it here to predict the values of the two components to observe their evolution in the future. Indeed, these represented 90.12% of our data (Figure 2). We preferred to reduce the information a bit but took two orthogonal variables (without correlations) representing all our data. Using the parameter “season = 12” gave us the best possible model. The *p*-values of 0.01 resulting from the Dickey–Fuller test indicated that our two time series were stationary and that we could integrate them into our model. The first step was to validate it to be sure of its stability, its fit, and its accuracy. To accomplish this, we carried a diagnosis of the residuals using several tests. First, the *p*-value < 0.001 indicated a good stability of the model. To test for serial correlation, we applied a Portmanteau test. The fact that the *p*-value was greater than 0.05 (0.73) indicated an absence of autocorrelation. We then performed a multivariate Lagrange multiplier ARCH test to test the absence of heteroscedasticity in the residuals, which proved convincing. We also performed Granger causality tests on the two components. The two *p*-values lower than 0.05 (0.005) suggested that the first component provided statistically significant information on the future values of the second component and vice versa. All these tests validated our model, and we could then proceed to the forecasts. First, forecasts for the test period (the last 26 months) of our data set were made based on the training period (first 130 months). The average error obtained from the forecasts was −3.1 with a 95% confidence interval of [−41.1; 34.8]. The average error was small, which showed us that our model was rather accurate. We could, therefore, make the forecast for the coming months. On the graph of Figure 5, the blue curve represents the forecast and the red lines are the 95% confidence interval. The prediction of the first component was close to the last recorded proportions. The annual periodic effect was found, and the model predicted neither an increase nor a decrease in the proportions of absences during the next 5 years. The second component with the lowest proportions of absences was expected to decrease slightly over the next 2 years while remaining periodic according to the VAR.

## 4. Discussion

The main findings were a higher proportion of absences for workers aged 30–50 years old (10.8% of contribution) with young children (6.4% of contribution), those who were tenured, and those in the administrative field (4.9% and 4.9%, respectively). The forecasts predicted a stagnation in the proportion of absences for the profiles of the most absent factors over the next 5 years including annual peaks.

### 4.1. Sociodemographic and Occupational Factors Influencing Absences

As stated earlier, we could not compare our results to other studies because none had been conducted over such a long period of time with such precision in the data. However, we could relate them to some analyses of hospital staff absences over shorter periods. First, the appearance of a high ratio of absences among women [25,26,27,28] since 2015 could be explained in part by the difficulty that mothers have in reconciling family life and professional life [29]. The relationship between age and absenteeism is a subject of great controversy. Indeed, we did not find a precise result; some found a correlation between old age and absenteeism [27] while others showed rather null or even curvilinear relations [30]. We found that people in couples were much more absent than single people, which confirms the results of the literature [31,32,33]. Similarly, being a parent influences absenteeism [34]. This is explained by financial obligations and family burdens as well as a conflict of roles as a parent and as a worker [30]. The high proportion of absences of caregivers [5,26,27] and tenured staff [27] was also confirmed due to their higher number of working hours. They are more prone to stress-related fatigue, particularly in the health care field where caregivers are under pressure from the challenges of their jobs [35]. The dissatisfaction and absenteeism of paramedical staff described in the literature [36] was only confirmed over the period 2011–2015 in our study. 

### 4.2. The VAR Model

We chose a VAR model to represent the evolution of our absences because it allowed us to obtain knowledge of future data for a time series according to the past data of this same series but also of several other series. This is a recent model that is not widely used in the literature but which proves to be a powerful and reliable tool for data description and forecasting [37]. It was perfectly adequate in our situation because we wanted to represent and predict the evolution over time of each variable according to the others. However, the VAR model could not consider a data set as large as ours. Indeed, after transformation, we had a table made of 50 correlated time series, which led to collinearity problems. For this reason, we had to perform a PCA beforehand, which acted as a variable selection. It allowed us to reduce the information of the 50 variables into only two main axes. Then, we were able to integrate these two axes into our VAR model. Instead of having the predictions of each variable, we obtained the predictions of profiles of variables.

### 4.3. Solutions to Reduce the Major Problem of Absenteeism

Some articles deal with solutions to understand and act on these absences [14,30,38,39]. In fact, there is still a lot of room to maneuver for companies, as 49% of those absent in 2019 believe that they could have implemented actions to avoid their absence [15]. Companies outside of the health care field have already decided to identify the causes of absenteeism and reduce them as much as possible thanks to action plans. These include self-replacement, adjustment of schedules, interdepartmental mobility, or the replacement pool [40]. Some companies set up meetings to quickly review the rate of absenteeism, monitor its evolution, carry out regular work and equipment changes to limit work-related accidents, or make schedule changes to reduce employee fatigue [38]. One study went further by focusing on the specific reasons for absences, the context surrounding them, and the potential impact they might have on the staff present [39]. The conclusion of the study explained that the reduction in absenteeism lies, above all, in the mutual trust between the manager and the employee and in the latter’s job satisfaction. Similarly, it has been found that the treatment of absenteeism requires an in-depth knowledge of employees (characteristics, expectations, motivations, etc.) and the implementation of programs that respond to their difficulties [14]. In the health care field, reductions in absenteeism have also been achieved through improvements in hospital management style and further research into absence behavior among these professionals [1,36].

## 5. Limitations

One of the problems we encountered in this study was missing data. For example, the distance variable had 231,140 missing data points. Moreover, one of the sites closed its doors in 2010; so, we did not consider its absenteeism data. Another did not record any absences in 2010 (geriatric and rehabilitation department). Another only opened in 2010 (secondary site) and, therefore, had no data before that date. This results in a missing data count of 222,283 for the establishment variable. These missing data points distorted our PCAs because these sites concentrated all the information on one axis. We then used the na.approx() function on R, which will perform a linear interpolation of the missing data according to the previous and following data. Concerning the secondary site, whose data only started in 2010, we proceeded to an automatic imputation with the function imputePCA(). Other articles have looked at different factors that we had not considered. Indeed, reforms, working hours, salary, stress, staff dissatisfaction, and seniority are variables that influence the rate of absenteeism [1,41,42,43]. It has been shown that the arduousness of the work as well as monotonous and repetitive work are factors influencing the number of absences [43]. In addition, staff satisfaction in the workplace plays an important role in the problem of absenteeism [42,43], and stress at work was said to cause half of all absences in 2010 [44]. It would then have been interesting to consider psychological variables such as the degree of satisfaction at work, the level of stress, the relationship with the personnel of the same service, or fatigue. Despite the imprecision of the data due to the self-assessment, the integrated variables could show important factors leading to high absenteeism. As mentioned earlier, absenteeism causes a financial problem for companies that must both pay the absent person and the person who will replace the vacant position plus the costs of managing the absence [45]. The cost of absenteeism in 2014 was estimated at EUR 60 billion for all types of absences combined [46]. The financial dimension of absenteeism is even a key indicator for companies. It would, therefore, be interesting in future studies to look at the salaries of absent people as well as the cost of replacing them.

## 6. Conclusions

The very high number of absences in our data set since 2007 revealed a serious problem in health care institutions. Absenteeism has been increasing significantly in this field for several years and can lead to medical concerns with a decrease in the quality of care, organizational problems, and financial concerns due to high replacement costs. Certain factors generate more absences than others, mainly professional factors, but it is also interesting to consider the sociodemographic variables of the absent persons. We deduced from our study the following profile of absentees: having a youngest child over the age of 4, being aged between 30 and 50 years old, working in the administrative field, being tenured, being a parent, and being in a couple’s relationship. Moreover, this profile remained almost the same throughout the study period. The predictions of the VAR model for the next 2 years showed an evolution similar to that of the last 30 months, with peaks each year for the variables representing the largest proportions of absences and a slight decrease for the variables with a lower percentage of absences.

## Figures and Tables

**Figure 1 ijerph-19-08236-f001:**
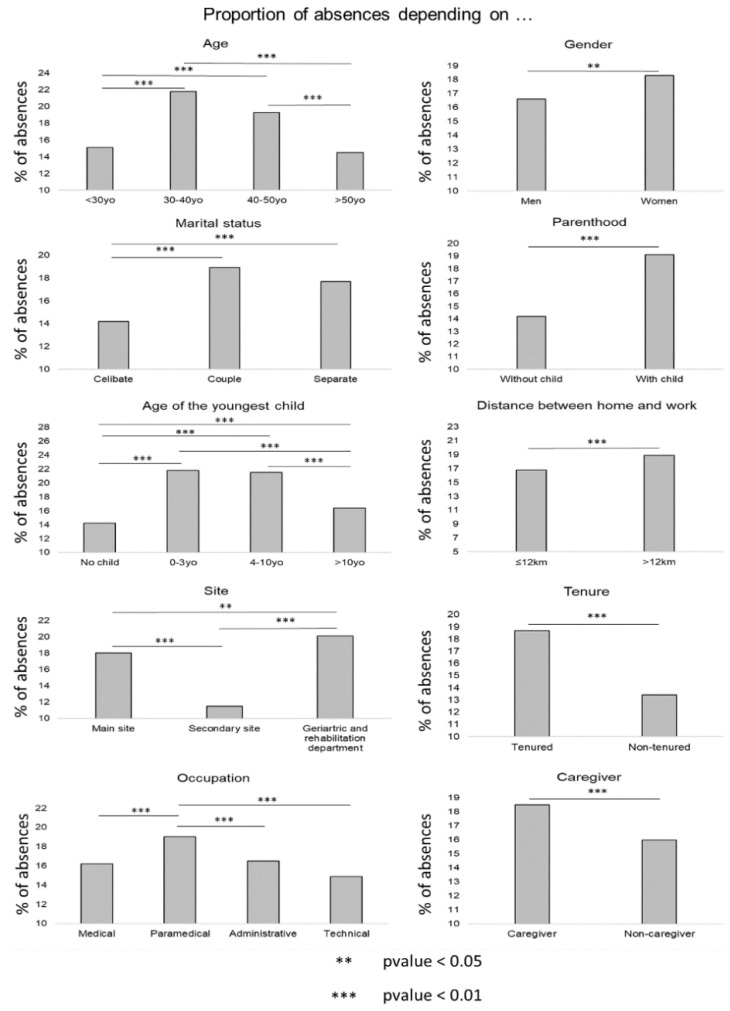
Proportion of absences depending on sociodemographic (age, gender, marital status, parenting, age of the youngest child, distance home/work) and professional variables (occupational groups, site, tenure, caregiver).

**Figure 2 ijerph-19-08236-f002:**
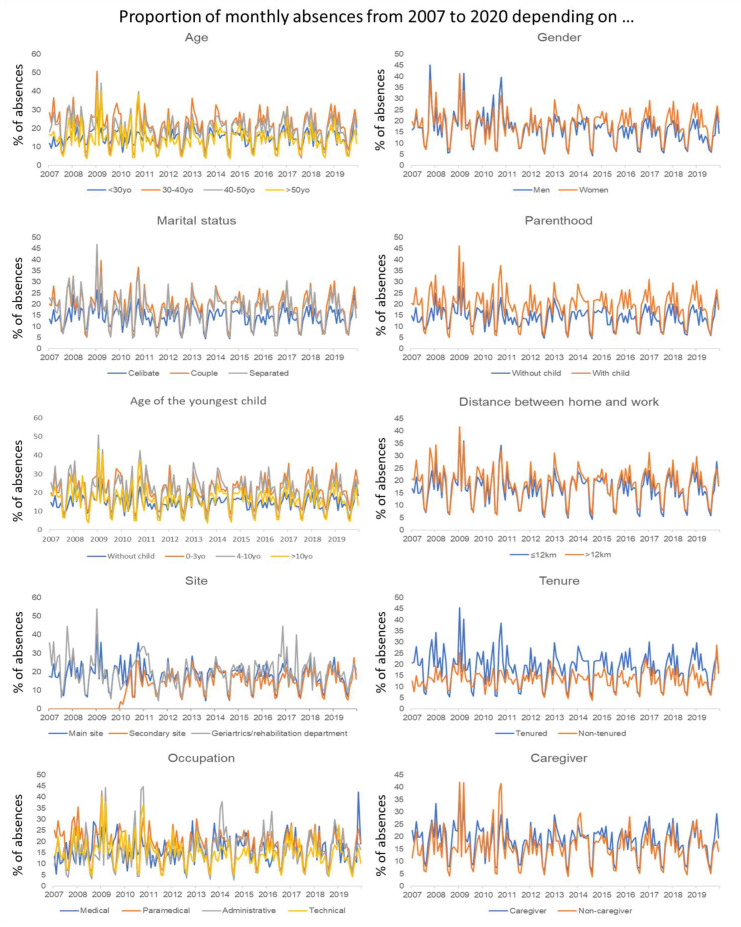
Proportion of monthly absences by sociodemographic and professional factors.

**Figure 3 ijerph-19-08236-f003:**
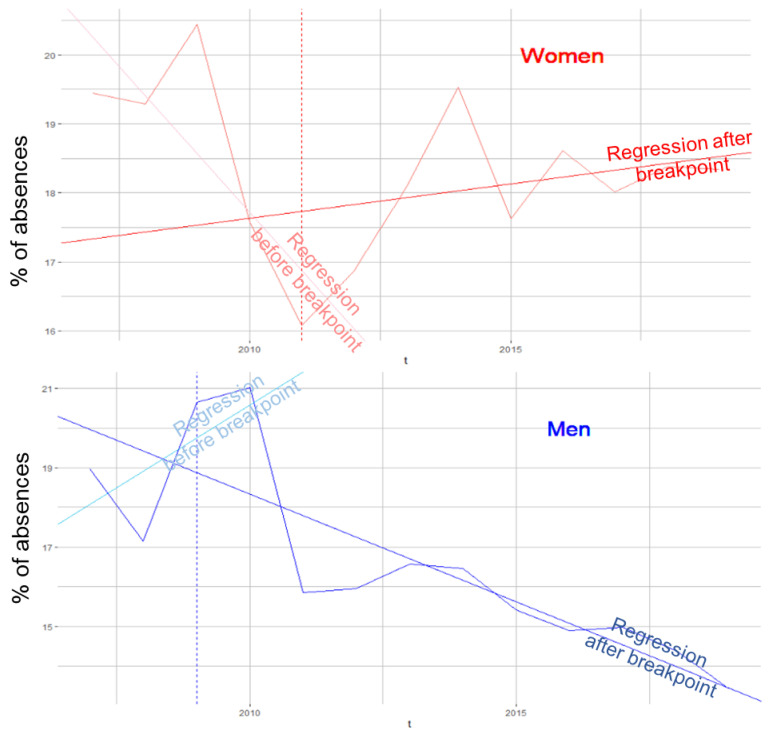
Proportion of annual absences by gender with linear trends and break points.

**Figure 4 ijerph-19-08236-f004:**
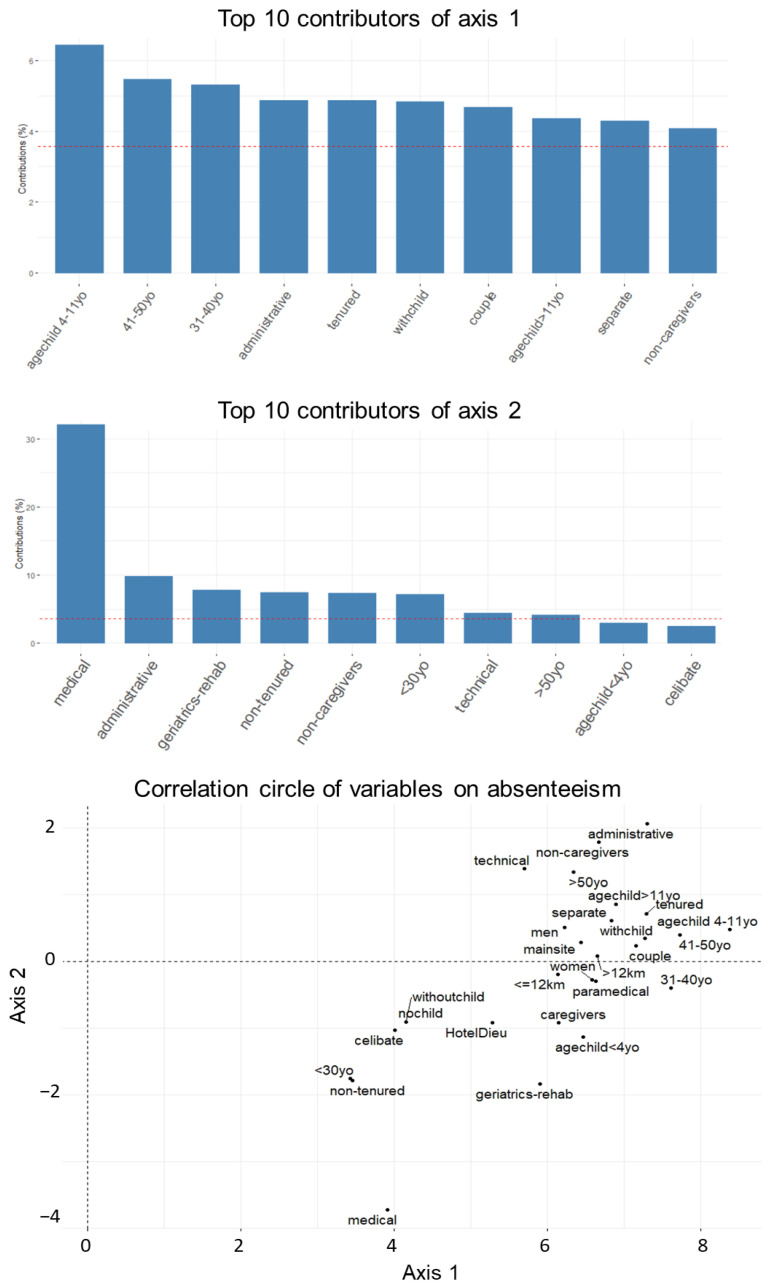
Largest contributors to the most absent (axis 1) and least absent (axis 2) profiles.

**Figure 5 ijerph-19-08236-f005:**
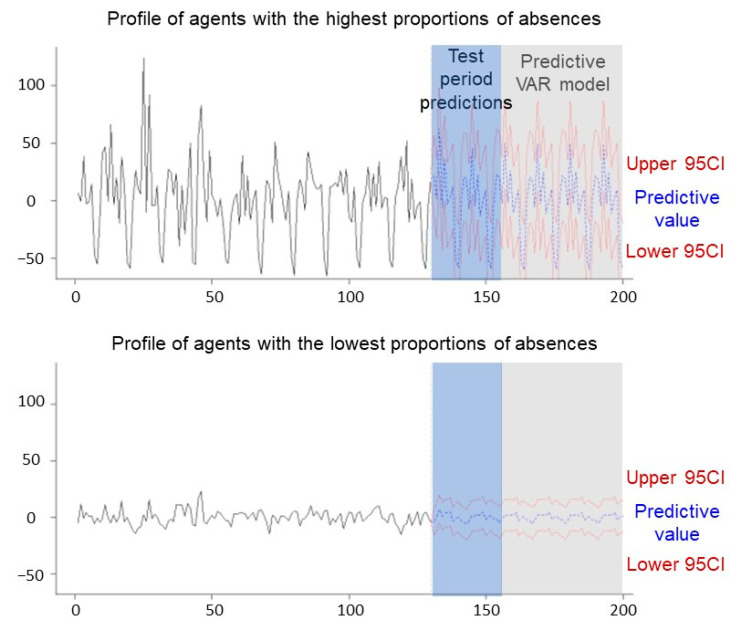
Evolution and predictions for the test period and for the months to come with the VAR model of the PCA components. The first component represents the factors of people with the highest proportions of absences, and the second component represents the factors of people with the lowest proportions of absences.

## Data Availability

Data cannot be shared publicly due to confidentiality. Data are available from the Institutional Data Access/Ethics Committee of the Clermont-Ferrand University Hospital (contact via the Human Resources Department) for researchers who meet the criteria for access to confidential data.

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
