# Peer review of "Mathematical Modeling of the Evolution of Absenteeism in a University Hospital over 12 Years"

_ijerph, 2022, doi:10.3390/ijerph19148236_

Round 1

Reviewer 1 Report

This manuscript analyzed the evolution of absences over time. The study leveraged a University Hospital data from 2007 to 2019 and performed advanced statistical modelling. The authors found that the variables contributing most to the typical profile of the highest proportions of absences were: having a younger child between 4 and 10 years old, aged between 40 and 50 years old, aged between 30 and 40 years old, work in the administrative field, be tenured, being a parent, be in a couple relationship, having a younger child over the age of 11, and be separated. These findings are meaningful, especially for targeted intervention for absenteeism.

I do believe, however, that a substantial revision in the method and result section will help make it a more valuable contribution to the literature. Below, I discuss a few major concerns, followed by a few minor points about the paper. Overall, I would suggest major revision.

Major comments:

1.     Line 131-132, “To do this, we performed a principal component analysis on our time series, treating months as individuals.42-43” Does you PCA account for the correlation between months?

2.     Line 139-141, “The na.approx() function in the zoo package on R was used to perform a linear interpolation of missing data for the geriatric and rehabilitation department in 2010.” What’s the missing proportion? What if you remove those missing records? I am wondering whether your conclusion is dependent on the missing value imputation procedure.

3.     Again, line 142-145, “For the secondary site with missing data from 2007 to 2010, the imputePCA() function in the missMDA package was more appropriate because we didn't just have one year without data in the middle of our dataset but 3 years of absences at the beginning.” What’s the missing proportion? What if you remove those missing records? I am wondering whether your conclusion is dependent on the missing value imputation procedure.

4.     Please separate your Statistics section into one section devoted to method and the other section devoted to statistical software used to implement the methods. Right now the presentation looks confused.

Minor comments:

1.     The first sentence of the paper: “Absenteeism is a very accurate barometer of the malfunctioning of an institution 40 and of the quality of working conditions.4” Why the paper started with the fourth reference? Please check your reference manager.

2.     Line 63, “particularly important in the health field,1-14-15-2-16” Is the format for the references (“1-14-15-2-16”) correct? Should it be “1,2,14,15,16”?

3.     Line 107-109, “Categorical parameters were expressed as frequencies and associated percentages, and continuous data as mean and standard deviation or as median and [25th; 75th percentiles], according to statistical distribution.” Could you elaborate on “according to statistical distribution”? In other words, how do you determine whether you use mean and standard deviation or quantiles?

4.     Line 118-119, “Then, with the help of student test, each modality was compared”. “student test” should be “student t test”.

5.     Figure 2: Please use the same background color for the 3 figures.

6.     Figure 4: Why there’s an info window for the figure on the upper left corner of the figure? How do you make the figure?

Reviewer 2 Report

The article addresses the important issue of absenteeism through a time-series study.

1. I think the authors should add some mathematical details of the methods used. They have used so many methods however it looks too congested without proper mathematical documentation. Even if all of that is not possible some minimal key methods should be mathematically presented

2. I found no summary plots of the initial analyses, at least how it varies over this 156 months across time. maybe they should add simultaneous plots of time series of different factors

3. As is natural for this type of time series, there is very significant amount of periodicity across years as there are exactly 12-13 peaks in each time series plot, how do they address this?

4. I am not sure if you just take every months as rows for the PCA purpose, as you suggested in the paper, are you not intentionally killing the dependence between them?

The VAR model came afterwards once this pca is extracted, but while doing this pca step you might have unintentionally killed a lot of information about the dependence

5. Finally the results are not surprising for the past data observed, there is not a very striking forecast for the next 5 years.. how do we justify significance of this analysis. The authors also only did this analysis on this data only so it is not clear whether their methodology has universal appeal
